# Drivers and Consequences of Short-Form Video (SFV) Addiction amongst Adolescents in China: Stress-Coping Theory Perspective

**DOI:** 10.3390/ijerph192114173

**Published:** 2022-10-29

**Authors:** Honglei Mu, Qiaojie Jiang, Jiang Xu, Sijing Chen

**Affiliations:** School of Economics and Management, Zhejiang University of Science and Technology, Hangzhou 310023, China

**Keywords:** short-form video (SFV) addiction, stress-coping theory, school burnout, social phobia, parental phubbing, wellbeing, happiness, parent–child relationship quality, perseverance

## Abstract

As a hedonic technology, short-form videos (SFVs) have attracted numerous users. However, one related problem that merits research is SFV addiction, especially among adolescents due to their immature self-control abilities. Although recent research has discussed the formation process of SFV addiction from psychological needs and the SFV features perspective, scarce attention has been paid to investigating the relationship between stress and SFV addiction, as well as the relationship between SFV addiction and its consequences. Therefore, the purpose of this study is to examine whether school burnout (school stress), social phobia (social stress), and parental phubbing (family stress) trigger SFV addiction based on stress-coping theory and, furthermore, whether SFV addiction leads to low levels of happiness (psychological consequence), parent–child relationship quality (relational consequence), and perseverance (behavioral consequence) among adolescents. The proposed model was tested based on data collected from 242 adolescents from across China under the age of 18 with the experience watching SFVs. A covariance-based structural equation modeling (CB-SEM) method was used for data analysis. The results showed that school burnout and social phobia significantly triggered SFV addiction, which later negatively and significantly influenced adolescents’ happiness, parent–child relationship quality, and perseverance. The study also found that SFV addiction served as a mediator between the drivers and consequences of SFV addiction. This study provides several theoretical implications. First, this study is one of the first to explain adolescents’ SFV addiction from stress-coping perspective, thereby enriching research in the field of SFV addiction. Second, prior research has rarely discussed the impacts of stresses from various environments on addiction behavior in a single study. Therefore, this study contributes to the knowledge of stress-related research in an SFV addiction context. Finally, our study enhances our understanding of the impact of SFV addiction on its consequences, in both an SFV research context and a social media research context.

## 1. Introduction

Short-form videos (SFVs), such as those found on TikTok, Instagram, Kuaishou, and Watermelon, are an emerging social medium, where users are allowed to create and make online SFVs via SFV apps [1]. Users can create videos that cover various aspects of their lives, e.g., cooking, songs and dances, fitness, pets, travel, health, and technology, ranging from seconds to minutes [2]. Users can also browse and like others’ videos to interact with them. The China Internet Network Information Center (CNNIC) reported that the number of SFV users in China was 888 million in 2021, up nearly 300 million from 2018, accounting for 87.8 percent of total Internet users.

SFVs provide personalized content for users based on an analysis of their preferences [2]. SFVs also offer fun stickers, video-editing tools, and special effect filters to help users create funny videos [2]. These customized contents and funny features have addictive hedonic value [2]. One hedonic-technology-usage-related problem that merits research is technology addiction [3]. For example, TikTok has 486 million monthly active users, and approximately 22 percent watch videos for more than 1 h per day [4,5]. This is especially true for adolescents due to their immature self-control abilities, along with the pursuit of psychological needs [6]. Excessive use may lead to a series of negative problems, such as increased depression and loneliness, distraction, low sleep quality, and social isolation [7,8,9]. However, compared to Internet addiction and social media addiction, SFV addiction has received far less scholarly attention [2,10].

By drawing on the definition of social media addiction, SFV addiction is defined as the extent to which a user attaches to SFVs strongly and cannot help using SFVs repeatedly, such that general addiction symptoms occur [11]. In line with common technology addiction, SFV addiction can be manifested through four key symptoms, including withdrawal (e.g., negative emotions arise if a user cannot use SFVs), salience (e.g., using SFVs dominates a user’s thoughts and tasks), conflict (e.g., using SFVs conflicts with other tasks), and relapse (e.g., SFV usage time cannot be reduced voluntarily by a user) [12]. There is no standard boundary to classify a user as addicted, and as such, most of the prior research has treated behavioral addiction as a continuous concept [3]. Continuous concepts assume that all users may have a certain level of addiction, ranging from low to high, although low for the majority of users [3]. This study adopts this perspective.

From stress-coping theory perspective, SFV addiction can be attributed to relieving negative emotions caused by stress in daily life. Stress-coping theory was first presented by [13] to understand how a user copes with a stressful condition and thereafter generates adaptation behavior. There are two continual stages: cognitive appraisals and coping processes. In cognitive appraisals, a user evaluates whether a specific environmental event poses potential consequences to himself or herself [14]. This stage can be further divided into primary and secondary appraisals. Through primary appraisal, a user mainly judges whether the potential consequences are positive or negative [14], while in the secondary appraisal, the user evaluates if any strategies can be applied to overcome the troubled person–environment relationship [14]. Coping processes mean specific behavioral and cognitive efforts to cope with a troubled person–environment relationship [15]. Users usually adopt situation-focused strategies and emotion-focused strategies to cope with environmental events. The former attempts to solve stressful situations proactively. The latter aims to reduce the negative emotional influence posed by stressful situations and increase the sense of well-being. SFV addicts are more likely to adopt emotion-focused strategies because their self-control and self-efficacy are at a relatively low level [16,17]. People with low self-control and low self-efficacy are more affected by stressful events and are more inclined to adopt emotion-focused strategies to cope with stress [18]. Therefore, this study attempts to discuss SFV addiction formation from an emotion-focused strategy perspective based on stress-coping theory.

Stress is an agitated state [19]. It occurs when there is a lack of approaches to satisfy the many environmental and social requirements, and it may generate a burden on one’s average ability to adapt [19]. Stress has been identified as a key predictive factor influencing behavioral addiction [10,19]. Groups tend to be addicted to smartphones when groups have more stress compared with groups that have less stress [20]. Prior research [19] found that stress significantly influenced adult smartphone addiction. Other research [9] confirmed that stress had a positive and significant impact on middle-aged adult SFV addiction. However, the question of whether stress leads to SFV addiction among adolescents is still unresolved. In addition, despite adolescents encountering stress from various environments, prior studies have consistently treated stress as a general construct. It is difficult to identify which stress triggers SFV addiction the most. As a student, one of the biggest stresses is school burnout, which is manifested in emotional states of pressure due to studying and feeling incompetent at one’s schoolwork. Furthermore, during the adolescent period, they experience a lot of stress from social phobia [21], which has a negative impact on social life. Another stress may come from family, which is parental phubbing [22]. Therefore, one of the purposes of this study is to examine whether school burnout (school stress), social phobia (social stress), and parental phubbing (family stress) trigger adolescent SFV addiction.

Rather than investigating only the drivers, this study is also interested in examining the consequences of SFV addiction. Prior studies have shown that social media addiction may cause psychological problems, including a low level of life satisfaction, loneliness [8], low happiness [23], and depression [24]. Nevertheless, scarce attention has been paid to the relational and behavioral consequences for adolescents. Adolescents spend more time at home with their parents, and this is especially the case during the COVID-19 pandemic since many classes were forced to switch to online. How SFV addiction influences the parent–child relationship quality when adolescents reduce the amount of time spent interacting with their parents needs further research. In addition, once the addictive behavior becomes a habit, this habit stimulates the use of SFV apps automatically whenever adolescents encounter difficulties, which may decrease their perseverance. Hence, another purpose is to investigate whether SFV addiction negatively impacts adolescents’ happiness (psychological consequence), parent–child relationship quality (relational consequence), and perseverance (behavioral consequence).

### 1.1. School Burnout and SFV Addiction

School burnout refers to a feeling of exhaustion due to study requirements, generating a detached attitude toward study and a feeling of incompetence as a student [25]. The feeling of exhaustion means an emotional state under pressure, and a detached attitude means a loss of interest in one’s study, while a feeling of incompetence means less successful achievement and accomplishment in one’s schoolwork [26]. Adolescents may easily experience school burnout when their study achievements failed to meet expectations [27], which may further lead to stress [28]. Stress is manifested in psychological states, such as overreaction, difficulty relaxing and feeling upset easily [10]. Adolescents who suffer from stress due to school are at high risk for smartphone addiction and online application addiction [29] because a stressful mental state can weaken adolescents’ self-efficacy and self-control abilities, which may decrease the capability of reducing excessive usage behavior automatically [16]. It is reasonable to argue that stress caused by academic performance increases adolescents’ school burnout, which results in a high possibility of using SFVs excessively [29]. Immersing themselves in SFVs is a coping strategy that can enable them to escape from school stress [29]. Stress was found to positively predict SFV addiction [10]. A prior study [29] also confirmed that adolescents who experienced school burnout tended to depend more on social media. Hence, we hypothesize that:

**Hypothesis** **1** **(H1).**
*School burnout positively affects SFV addiction.*


### 1.2. Social Phobia and SFV Addiction

Social phobia refers to the fear of being evaluated by others and an emotional state of feeling embarrassment, abasement, or being afraid of becoming a laughingstock due to behavior [21,30]. Such a negative mental state becomes a barrier for adolescents to enter into social surroundings and establish social relationships [30]. It is worth mentioning that social phobia is different from social interaction anxiety, a construct that has been widely investigated in Internet addiction and social media addiction contexts [2]. Social interaction anxiety reflects a general level of anxiety associated with the beginning and maintenance of social interactions, such as meeting and talking to friends or strangers, making eye contact with others, etc. [31]. However, social phobia reflects a level of experienced anxiety associated with completing various tasks while being watched by others [31]. The state of social phobia may appear without interacting with a specific person. There are many events that can cause adolescent social phobia, including being stared at by strangers while walking down the street, sitting facing strangers on a subway or train, and being afraid to attract the attention of other people [31]. Given that social phobia receives little attention in the behavioral addiction context, this study aims to explore the relationship between social phobia and SFV addiction.

Social phobia impairs the quality of adolescents’ social life and prompts them to use the Internet as a low-risk alternative [21]. The social compensation assumption states that people with high levels of social problems are tempted to seek out social media [32]. As a type of social media, SFV apps provide enjoyable videos and comfortable virtual social environments for adolescents, which help them to cope with stress caused by social phobia [21]. This may result in a cycle of avoidance that is perpetuated by watching SFVs and forms the addictive behavior of using SFVs. A prior study found a positive relationship between social phobia and Internet addiction [21]. Thus, following a similar vein:

**Hypothesis** **2** **(H2).**
*Social phobia positively affects SFV addiction.*


### 1.3. Parental Phubbing and SFV Addiction

Parental phubbing refers to a parent’s act of snubbing their children in daily parent–child conversations or in daily time by focusing on a mobile phone [33]. On the one hand, during a parent–child conversation, parental phubbing behavior may distance the intimate relationship with their children and make them feel they are excluded [6]. From the children’s perspective, they perceive that mobile phones are more vital than they are [34]. On the other hand, parental phubbing in daily life may set a “bad example” of using mobile phones excessively, accelerating children’s addictive behavior to mobile phones [35]. In either case, it can exacerbate the pressure on adolescents and push them toward SFV apps. According to parental acceptance–rejection theory [36], parents’ neglect and ignorance may impair adolescents’ self-esteem, and increase anxiety, depression, and other well-being problems [37,38]. These maladaptive psychological states may push adolescents to use SFVs in order to escape from reality temporarily [6]. The entertainment and personalization features of SFVs attract adolescents to repeat their enjoyable actions, which can eventually form a habit [12]. Once a habit is formed, using SFVs is performed automatically every time they encounter parental phubbing behavior, leading to addiction behavior eventually. Research on Internet addiction [22,39] has consistently discovered that parental phubbing positively and significantly impacts Internet addiction. In addition, parental phubbing was found to be the core determinant of SFV addiction among Chinese adolescents [6]. Therefore, this study presents the following hypothesis:

**Hypothesis** **3** **(H3).**
*Parental phubbing positively affects SFV addiction.*


### 1.4. SFV Addiction and Happiness

Personal happiness is one dimension of psychological well-being and is treated as the most important goal in our life [40]. It is a fundamental demand for human beings and is relevant for the judgment of a good life [41]. Happiness is defined as a state including a high level of satisfaction toward life and frequent positive emotions, as well as infrequent negative emotions [42]. Happiness is closely related to our life, and many factors may affect our happiness. It was argued that, in the current society where social media are embedded in adolescents’ daily lives, excessive social media use may pose threats to happiness [43]. According to [41], adolescents who used social media excessively or never used social media all reported lower levels of happiness than those who used it moderately. Excessive social media usage led to lower happiness. However, some studies [44] did not find any associations between overall screen time and well-being across adolescents whose screen time was below or above certain thresholds. Obviously, the proposition that excessive social media usage is detrimental to happiness has received inconsistent support. The fun stickers, special effect filters, and video-editing tools afforded by SFVs help adolescents create funny videos. The personalized recommendation functions based on analyses of adolescents’ preferences may tempt them to consume SFVs excessively and become SFV-addicted [2,10]. High amounts of SFV usage dominate adolescents’ everyday life, weaken their self-control ability over behaviors and thoughts, distract their attention from focusing on schoolwork, and decrease their academic performance, which may be harmful to happiness. Therefore, we hypothesize that:

**Hypothesis** **4** **(H4).**
*SFV addiction negatively affects happiness.*


### 1.5. SFV Addiction and Parent–Child Relationship Quality

The parent–child relationship quality concept was proposed as early as the 1850s [45], referring to the relationships between parents and their children, including biological, adopted, and stepchildren. A positive and healthy parent–child relationship quality is beneficial to protecting adolescents’ mental health, risk-taking, and behavioral problems [46,47,48]. It has been well-established that social media addiction may be harmful to adolescents’ well-being, academic performance, sleep quality, and healthy social relationships [8,43]. Nevertheless, there is a paucity of studies that investigate the association between social media addiction, particularly SFV addiction, and parent–child relationship quality. Although few studies have examined the impact of SFV addiction on parent–child relationship quality, insight can be gleaned from Internet-use-related research. Prior studies have found that Internet use [49] and technology use [50] both negatively influence parent–child relationship quality, and playing online games decreased communication time with parents [49]. This also holds true in an SFV context. On the one hand, using SFVs excessively reduces the interaction time with parents. On the other hand, spending too much time on SFVs can cause concerns for parents, such as impacting vision and reducing academic performance. Parents prevent adolescents from immersing themselves in SFVs, but there may be arguments and conflicts during prevention, which may hamper the parent–child relationship quality. Thus, based on prior research, this study hypothesizes that:

**Hypothesis** **5** **(H5).**
*SFV addiction negatively affects parent–child relationship quality.*


### 1.6. SFV Addiction and Perseverance

Perseverance is one facet of non-cognitive skill grit, which is defined as perseverance to complete long-term goals under challenges and setbacks [51]. Therefore, it consists of “consistency of interests over time” and “perseverance of effort”. For the purpose of this study, we intend to explore the impact of SFV addiction on adolescents’ perseverance. In this study, perseverance refers to adolescents’ decisions to persevere in tasks they consider relatively difficult [51]. The necessary presuppositions of perseverance are self-regulation, or self-control [17,52], meaning an adolescent’s capability to inhibit desires that stop them from recognizing self-regulatory goals [53]. According to distraction-conflict theory (DCT) [54], as a potential distraction mechanism, the ease of access and entertainment features of social media can easily distract users from their primary tasks [23]. Social media poses challenges to users’ self-control because they are in a dilemma to make a decision between tempting entertainment and long-term goals and obligations [53]. Users who engage in social media excessively often fail to regulate or control their usage behaviors [55], indicating a lack of perseverance to override difficulties in tasks. Similarly, adolescents who are addicted to SFVs are immature in regulating and controlling themselves [6]. The enjoyable SFVs dominate their minds and make them unable to focus on their schoolwork. Adolescents may escape from difficulties and challenges temporarily by switching to SFVs. Thus:

**Hypothesis** **6** **(H6).**
*SFV addiction negatively affects perseverance.*


### 1.7. SFV Addiction as a Mediator

Similar to social media addiction, SFV addiction is another subcategory of Internet addiction [2]. School burnout has been found to increase the risk of being addicted to social media [29]. Adolescents with high school burnout immersed themselves in SFVs to escape from school stress [29]. Another stress comes from social phobia. Social phobia was found to have a positive effect on Internet addiction [21]. Adolescents with high social phobia attempt to seek out social media to cope with stress caused by social phobia [32]. The entertainment and personalization features of SFVs help adolescents avoid ongoing embarrassment and nervous feelings, which may create addictive tendencies. As discussed above, parental phubbing was found to be another core determinant of SFV addiction [6]. Parental phubbing behavior may make adolescents feel they are excluded and neglected and, in turn, impair their self-esteem and increase their depression [6]. These maladaptive psychological states may eventually push adolescents to seek out SFVs to escape from the stress of their families. Adolescents with high levels of school burnout, social phobia, and parental phubbing increase their possibility of becoming maladaptively dependent on SFVs, which may dominate adolescents’ thoughts and schoolwork, raise negative emotions if they cannot use SFVs, conflict with other tasks, and make them unable to reduce SFV usage voluntarily. The abovementioned addiction symptoms may further reduce adolescents’ happiness, hamper parent–child relationship quality, and pose a challenge to adolescents’ self-control ability. Prior studies have confirmed that excessive social media usage can reduce happiness, parent–child relationship quality, and perseverance [23,50,53]. Therefore, this study proposes the following hypotheses:

**Hypothesis** **7** **(H7).**
*SFV addiction mediates the effects of school burnout on happiness.*


**Hypothesis** **8** **(H8).**
*SFV addiction mediates the effects of school burnout on parent–child relationship quality.*


**Hypothesis** **9** **(H9).**
*SFV addiction mediates the effects of school burnout on perseverance.*


**Hypothesis** **10** **(H10).**
*SFV addiction mediates the effects of social phobia on happiness.*


**Hypothesis** **11** **(H11).**
*SFV addiction mediates the effects of social phobia on parent–child relationship quality.*


**Hypothesis** **12** **(H12).**
*SFV addiction mediates the effects of social phobia on perseverance.*


**Hypothesis** **13** **(H13).**
*SFV addiction mediates the effects of parental phubbing on happiness.*


**Hypothesis** **14** **(H14).**
*SFV addiction mediates the effects of parental phubbing on parent–child relationship quality.*


**Hypothesis** **15** **(H15).**
*SFV addiction mediates the effects of parental phubbing on perseverance.*


### 1.8. Control Variables: Gender, Grade, and Watching Time per Day

In the SFV addiction context, watching time (duration of use, alternatively) is regarded as an important factor in regulating addictive behavior [10]. In the current study, adolescents’ watching time per day may impact their SFV addiction behavior, as well as the consequences of SFV addiction. In addition, prior research also suggested that demographic variables, such as gender and grade, influenced social media usage behavior [56]. Therefore, this study included gender, grade, and watching time per day as control variables in the research model. Figure 1 shows the research model.

## 2. Materials and Methods

### 2.1. Data Collection and Participants

This study adopted an online survey method to collect data. The questionnaire was made through Sojump (Wenjuanxing), a professional online questionnaire survey, examination, evaluation, and voting platform. Sojump has 2.6 million registered respondents with various demographic characteristics distributed across China. Sojump has been widely adopted to collect survey data under different research contexts, including e-commerce, air pollution, renewable energy, and behavioral addiction [10,57]. Sojump supports a variety of screening rules for invalid questionnaires. For example, it can set a too small amount of time to screen for randomly filled answers. Thus, this study used Sojump to collect data.

The target respondents were adolescents under the age of 18 who had experience watching SFVs. The current research was approved by the Ethics Committee of our university and informed consent from adolescents and parents was obtained. Determining the appropriate sample size was important for ensuring the quality of the study. Therefore, the sample size and anticipated effect were articulated prior to the completion of the survey instrument. According to [58], the sample size for this study could be reached at least one hundred and seventy participants. The questionnaire was randomly sent out and was open for one month in August 2022. A total of two hundred and sixty-four adolescents participated in the survey. Twenty-two were removed due to (1) the adolescents having no experience using SFV apps or (2) the answers given by the adolescents being the same. This resulted in two hundred and forty-two valid data. Among them, 83 (34%) were boys, and 159 (66%) were girls. Most adolescents were in high school (132–55%), and almost 62 (26%) were in junior school. Furthermore, 103 (43%) watched SFVs for one to two hours, and 54 (22%) watched for two to four hours per day. A total of 18 (7.4%) adolescents watched SFVs for four to six hours, and 9 (3.7%) of them even watched for more than six hours per day. According to [2,5], using SFV apps for more than 1 h per day was a certain level of SFV addiction. Our survey results indicated that most adolescents were actually addicted to SFVs. Moreover, approximately 152 (63%) preferred Kuaishou best, followed by the TikTok (147, 61%) and Sina Weibo (56, 23%) platform. Table 1 presents the descriptive statistics of the adolescents.

### 2.2. Measurement

Initially, 25 items were adapted and modified from prior studies using a five-point Likert scale, with 1 meaning “strongly disagree” and 5 meaning “strongly agree”. For instance, four items were adopted from [26] to measure school burnout. However, the first item was removed due to low factor loading (less than 0.5). Social phobia included four items that were adapted from [31]. The factor loading of the first item was lower than 0.5 and, thus, was deleted finally. Parental phubbing was measured using three items adapted from [59]. SFV addiction was measured using four items that were adapted from [12] and [60]. Happiness involved three items that were adopted from [42]. Different from ranging from “1 = strongly disagree” to “5 = strongly agree”, the first item of happiness ranged from “1 = very not happy” to “5 = very happy.” The second item of happiness ranged from “1 = strong dissatisfaction” to “5 = strong satisfaction.” Four items were employed from [61] and [45] to measure parent–child relationship quality. It is worth mentioning that items 2, 3, and 4 of parent–child relationship quality were reverse-coded. Finally, perseverance was measured using three items employed from [62]. The measurement items and sources are shown in Appendix A in detail.

## 3. Results

The current study employed a covariance-based structural equation modeling (CB-SEM) method using AMOS (version 23.0) software to test our measurement and structural model. AMOS was suitable for this study because it provided multiple goodness-of-fit indices [63] and could assess the direct influences of the drivers on SFV addiction while testing the consequences of SFV addiction simultaneously. At the same time, it could also measure the indirect effects of the drivers on the consequences through SFV addiction. Thus, this study used AMOS 23.0 software for data analysis.

### 3.1. Measurement Model

First of all, the measurement items for parent–child relationship quality questions 2, 3, and 4 were reversed before analysis. The measurement model was tested through confirmatory factor analysis (CFA). As shown in Table 2, the normalized chi-square (χ^2^/df) was 1.584, the comparative fit index (CFI) was 0.926, the incremental fit index (IFI) was 0.928, the adjusted goodness-of-fit index (AGFI) was 0.892, and the root mean square error of approximation (RMSEA) was 0.049. All the values satisfied the threshold, suggesting a good model fit.

Cronbach’s alpha was mainly used to test the model’s reliability. As shown in Table 3, the values were greater than the threshold of 0.7 [64], indicating good reliability. Convergent validity should gratify three requirements: factor loadings higher than 0.7, composite reliability (C.R.) greater than 0.7, and average variance extracted (AVE) higher than 0.5 [65]. Table 4 shows that all the item loadings not only surpassed 0.7, but also exceeded the cross-loadings [64]. Table 3 also presents the values of C.R. and AVE. Both values exceeded the accepted levels. These results indicated that the data had a good level of convergent validity. Then, this study tested the discriminant validity. Table 3 shows that the square root of the AVE was greater than the correlation coefficients of the interconstruct, indicating good discriminant validity.

This study adopted a self-report method to collect data, and as such, common method bias (CMB) could become a potential problem. To assess if CMB contributed to the variance among the survey items significantly, this study used two methods. First, Harman’s single-factor [66] test was conducted. Seven factors were generated, and the largest factor captured 25% of the variance. Second, following [67], a common method variable (CMV) was added to the research model. As shown in Table 5, the average substantive factor loading was 0.663, while the average method-based factor loading was 0.005. Both analyses demonstrated that CMB was not a concern for our data.

### 3.2. Structural Model

Afterward, a structural model was assessed. All the values of the model fit indices fitted well with the data (χ^2^/df was 1.838, CFI was 0.900, IFI was 0.902, AGFI was 0.870, and RMSEA was 0.059). Figure 2 shows the results of the hypotheses. All the hypotheses were supported, except for H3. Among the drivers of SFV addiction, social phobia exerted the largest impact (β = 0.353, *p* < 0.01), followed by school burnout (β = 0.211, *p* < 0.05). However, parental phubbing failed to influence SFV addiction significantly (β = 0.043, *p* > 0.05). In terms of consequences, the results indicated that SFV addiction had negative and significant effects on adolescents’ happiness (β = −0.369, *p* < 0.001), parent–child relationship quality (β = −0.469, *p* < 0.001), and perseverance (β = −0.170, *p* < 0.05). The control variables of gender (β = −0.296, *p* < 0.05) and watching time per day (β = 0.292, *p* < 0.001) also influenced SFV addiction significantly. The control variable of grade failed to impact SFV addiction significantly. In addition, our results showed that the three control variables of gender, grade, and watching time per day had no significant impact on happiness, parent–child relationship quality, and perseverance, respectively. The main constructs and control variables of this study jointly explained 39.8% of the variance in SFV addiction. SFV addiction and control variables, in turn, explained 16.8% of the variance in happiness, 14.1% of the variance in parent–child relationship quality, and 9.2% of the variance in perseverance.

The mediation effects were tested by following the formula used by [68]. The indirect effects of the drivers of SFV addiction on the consequences of SFV addiction were calculated. According to [68], a mediating effect exists if the *p*-value is significant. Afterward, both the lower bound and upper bound values are analyzed. Indirect effects can be confirmed if the lower bound and upper bound values do not have zero between them with a confidence interval (CI) of 95%. As shown in Table 6, the indirect effects of school burnout (*p* < 0.05, LL = −0.241, UL = −0.001) and social phobia (*p* < 0.05, LL = −0.295, UL = −0.028) on happiness were significant; thus, H7 and H8 were accepted. The indirect effects of school burnout (*p* < 0.05, LL = −0.294, UL = −0.002) and social phobia (*p* < 0.05, LL = −0.378, UL = −0.022) on parent–child relationship quality was significant, indicating that H10 and H11 were supported. The indirect effects of social phobia (*p* < 0.05, LL = −0.193, UL = −0.003) on perseverance was significant; thus, H14 was accepted. However, the indirect influence of parental phubbing on happiness (*p* > 0.05, LL = −0.129, UL = 0.069), parent–child relationship quality (*p* > 0.05, LL = −0.168, UL = 0.079), and perseverance (*p* > 0.05, LL = −0.074, UL = 0.027) were insignificant; thus, H9, H12, and H15 were rejected. In addition, the indirect effect of school burnout on perseverance (*p* > 0.05, LL = −0.177, UL = 0.000) was not significant; thus, H13 was also rejected.

## 4. Discussion

The current study attempted to explore whether school burnout, social phobia, and parental phubbing triggered SFV addiction and, simultaneously, whether SFV addiction decreased the levels of happiness, parent–child relationship quality, and perseverance among adolescent SFV users. A research model was proposed based on the literature and was tested through a survey of 242 adolescent SFV users. This study drew some important findings. First, stress from social phobia was identified as the most significant driver that positively impacted SFV addiction. This result is consistent with [22], which revealed that adolescents with social phobia spent more time on the Internet and were Internet addicts. Prior research on SFV addiction investigated how social interaction anxiety affected SFV addiction [2], but this is different. Social interaction anxiety is a problematic disorder when communicating with others face-to-face, whereas social phobia is a problematic disorder without face-to-face communication with others [31]. Adolescents may experience more stress from social phobia because social phobia exists even when there is no direct interaction with others. This study proved that social phobia is a central indicator of SFV addiction. Immersing themselves in SFVs can become a coping strategy for adolescents to avoid the embarrassments caused by social phobia. Such automatic usage can gradually become SFV addiction behavior.

Second, this study found that school burnout was another predictor of SFV addiction. This finding is in line with recent research showing that school burnout directly impacted disturbed sleep due to social media use [29]. Low academic achievement and excessive schoolwork can result in higher school burnout levels, which in turn, increase the possibility of using SFVs as a coping strategy, leading to higher SFV use and becoming addicted to SFVs gradually. The finding supports the idea that immersing oneself in SFVs is a coping strategy for adolescents to escape from the school context [39]. It could be concluded that, the higher the level of school burnout, the more likely adolescents are to be addicted to SFV apps.

Third, contrary to the study of [6], this study revealed that parental phubbing was not a significant driver of SFV addiction. A plausible explanation is that the behavior of holding a mobile phone has become a habit, no matter for parents or adolescents. Therefore, adolescents may understand parental phubbing behavior and not feel ignored by their parents. Another possible reason is that adolescents’ SFV addiction behavior is formed by the attractive features of SFV apps [2] and, simultaneously, by the stress from the school and social contexts. The mechanism of online and offline interaction results in SFV addiction and, thus, is less affected by parental phubbing. A third possible explanation is that the large and significant influences of school burnout and social phobia weakened the impact of parental phubbing.

Fourth, this study found that SFV addiction harmed adolescents’ happiness, which is in line with [23], who suggested that social media use lowered the level of happiness. One of the symptoms of SFV addiction is hardly reducing usage time consciously. DCT [55] states that, when a user is distracted by a hedonic IT (e.g., SFV apps), they may forget to process primary tasks, and some cues may have difficulty entering a user’s memory again. When they intend to complete primary tasks, a recovery period is needed to reprocess the unfinished primary tasks. Due to SFV addiction, adolescents may encounter difficulties finishing schoolwork on time, and attractive SFVs may dominate adolescents’ thoughts when they attempt to focus on schoolwork. These issues can lower their happiness.

Fifth, the results also indicated a negative relationship between SFV addiction and parent–child relationship quality. This finding is in congruence with [47], who found that excessive use of social media was negatively associated with parent–child relationship quality. As one type of social media, heavy SFV app usage could reduce interaction time with parents and impinge upon relationship quality with parents [47]. Immersing themselves in SFV apps may cause conflicts and disputes with parents because parents generally prevent their children from using SFV apps, which may lead to quarrels and further hamper parent–child relationship quality.

Sixth, this study found that SFV addiction negatively influenced adolescents’ perseverance. We intend to provide a plausible explanation based on DCT theory [55]. This theory states that hedonic IT (e.g., SFV apps) can distract users’ attention and lower their self-control. If a user perceives a particular social networking site (SNS) as central to who he or she is and has a strong IT identity with that SNS, it decreases his or her self-control over time [17]. Self-control is one necessary presupposition of perseverance. A lack of self-control may further indicate a lack of perseverance to overcome difficulties in tasks. Our results support this hypothesis.

Finally, SFV addiction appeared to mediate the relationship between school burnout and happiness, as well as parent–child relationship quality. This implies that the more school burnout adolescents have, the more SFV addiction has a negative influence on their happiness and parent–child relationship quality. However, SFV addiction failed to mediate the relationship between school burnout and perseverance. A plausible reason could be that perseverance was more impacted by social phobia through SFV addiction (*p* < 0.01, LL = −0.193, UL = −0.003), thus weakening the indirect influence of school burnout through SFV addiction (*p* > 0.05, LL = −0.177, UL = 0.000). Social phobia could negatively impact happiness, parent–child relationship quality, and perseverance through SFV addiction. The results indicated that the more social phobia adolescents had, the more SFV addiction had a negative impact on their happiness, parent–child relationship quality, and perseverance. Since parental phubbing failed to significantly influence SFV addiction, the mediation relationships between parental phubbing and happiness, parent–child relationship quality, and perseverance also failed. The above results underlined the roles of school burnout and social phobia as vital risk factors for adolescents’ happiness, parent–child relationship quality, and perseverance [21,29,50].

### 4.1. Theoretical Implications

This study provided several theoretical implications. First, our study introduced stress-coping theory into the SFV addiction context. Prior research has applied the opponent process theory [5], life history theory [69], and a combination of the attachment theory with a socio-technical approach [2] to study SFV addiction, but it has seldom considered how stress impacts SFV addiction. This study is one of the first to explain adolescents’ SFV addiction from stress-coping perspective, thereby enriching research in the field of SFV addiction.

Second, despite past evidence suggesting that stress [10], parental phubbing [6], and social interaction anxiety [2] are predictors of SFV addiction, researchers have rarely discussed the impacts of stresses from different environments on addictive behavior in a single study. Hence, we know little about which stress adolescents encounter the most and which stresses are more likely to lead to SFV addiction. Our study considered adolescents’ stresses from various contexts comprehensively, including stress from the school context, the social context, and the family context. Our results indicated that stress from social sources was the most salient potential factor leading to SFV addiction. Therefore, this study contributed to the knowledge of stress-related research in the SFV addiction context.

Third, this study is among the first intended to explore the consequences of SFV addiction. Many studies have investigated the formation mechanism of SFV addiction, but the relationships between SFV addiction and adolescents’ psychological, relational, and behavioral consequences are still unclear. Although social-media-addiction-related research has widely discussed adolescents’ well-being, very few empirical studies have focused on adolescents’ personalities and their relationships with their parents. Our results suggested that SFV addiction had a great negative impact on adolescents’ happiness and parent–child relationship quality, as well as perseverance. Thus, our study enhanced our understanding of the impacts of SFV addiction on its consequences, in both an SFV research context and a social media research context.

### 4.2. Practical Implications

This study also provided several practical implications. First, social phobia was a key predictor of SFV addiction. On the one hand, schools are suggested to organize extracurricular activities, such as sports, to build real relationships among adolescents. On the other hand, parents should organize family gatherings regularly. Stable relationships with classmates and family can improve adolescents’ social skills, enhance their confidence and intimacy when facing other people, and expand their social networks, thereby reducing social phobia. Therefore, schools and parents should work together to reduce adolescents’ social phobia.

Second, schools and parents should be cautious that school burnout can push adolescents to SFV apps and form addiction behavior gradually. School burnout can be attributed to low academic achievement, excessive schoolwork, and low confidence in learning ability. In this regard, it is recommended for schools reduce the quantity of schoolwork appropriately. Both schools and parents should take into account potential threats of low academic achievement and encourage adolescents whose academic performance is decreasing to enhance their self-confidence.

Third, our results confirmed that SFV addiction could lower the level of happiness, negatively influence parent–child relationship quality, and decrease adolescents’ perseverance. Several suggestions can be provided to parents and schools. First, parents are advised to discuss the dark side of SFVs with adolescents and establish reasonable rules concerning time spent on SFV apps. Schools can educate adolescents about the potential negative influence of excessive SFV use. Second, another effective method to reduce the usage of SFVs is switching adolescents’ attention by providing alternative activities, including hobbies, family gatherings, travels, parties with classmates, etc. Last but not least, activating a busy mindset may be a more effective nudge to control excessive SFV app use. A busy mindset refers to one’s subjective perception of having a lot of work or tasks to do rather than a perception of busyness due to deadlines or time pressure [70]. It was confirmed that a busy mindset is a sign of a good life and can facilitate self-control behavior [70].

### 4.3. Limitations and Future Research

Despite the implications, this study has several limitations that can be considered in future studies. First, though our sample was collected across China, a sample of two hundred and forty-two was not representative enough on a national scale. Future research is suggested to expand the scope of sampling and target more adolescents. Second, the findings of this study may not be generalized to other cultural contexts since school burnout, SFV apps, and parent–child relationship quality are culturally dependent. Future studies can conduct cross-cultural research to test the research model. Third, this study only determined three stress factors and three consequences. Future studies can explore more stress indicators from various environments and more outcomes related to well-being and personality. Finally, this study did not investigate prevention and SFVs addiction reduction factors. Future studies are recommended to include moderators, such as a busy mindset, in the research model. In addition, personality may serve as a moderator among adolescents’ SFV addiction behavior, which requires future studies to explore.

## 5. Conclusions

SFV addiction is a pathological factor that negatively influences adolescents’ mental states. This study attempted to examine whether SFV addiction was a stress-coping strategy result by identifying different stresses, including school burnout, social phobia, and parental phubbing, which have not been investigated in a single research effort. Furthermore, the current study also aimed to explore the relationships between SFV addiction and adolescents’ happiness, parent–child relationship quality, and perseverance. Our results demonstrated that SFV addiction was triggered by social phobia and school burnout. The results also found that SFV addiction had negative impacts on happiness, parent–child relationship quality, and perseverance. These findings not only contribute to SFV addiction and social media addiction research but also to schools and parents taking measures to prevent (potential) SFV addiction behavior.

## Figures and Tables

**Figure 1 ijerph-19-14173-f001:**
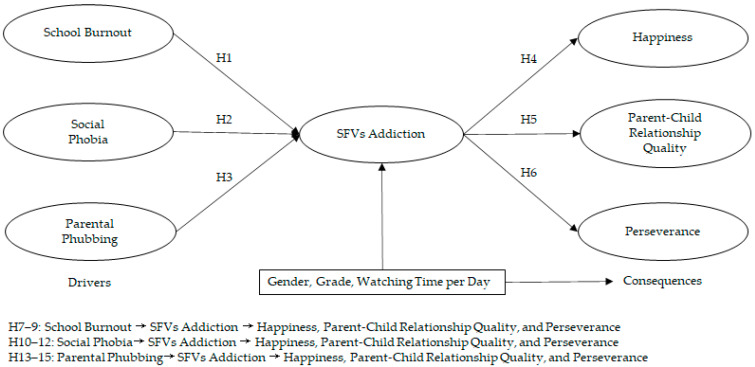
The research model.

**Figure 2 ijerph-19-14173-f002:**
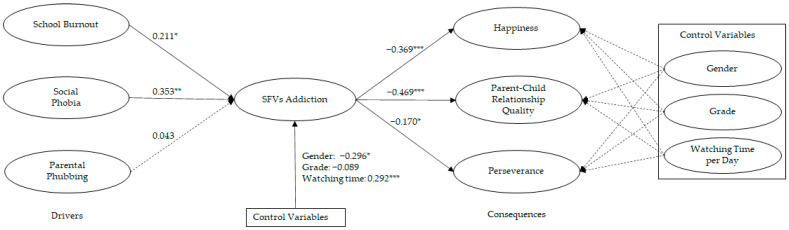
Model test results. Note: * *p* < 0.05; ** *p* < 0.01; *** *p* < 0.001.

**Table 1 ijerph-19-14173-t001:** Demographic characteristics of the participants (*n* = 242).

Measure	Item	Frequency	Percentage (%)
Gender	Boy	83	34.3
	Girl	159	65.7
Grade	Primary school	14	5.8
	Junior school	62	25.6
	High school	132	54.5
	University	34	14
Watching time per day	Less than 1 h	58	24
	1–2 h	103	42.6
	2–4 h	54	22.3
	4–6 h	18	7.4
	More than 6 h	9	3.7
Which short-form video app do you use often? (Multiple choice)	TikTok	147	60.7
	Kuaishou	152	62.8
	Sina Weibo	56	23.1
	Watermelon	8	3.3
	Tencent	42	17.4
	Other	60	24.8

**Table 2 ijerph-19-14173-t002:** Model fit indices.

Fit Indices	χ^2^/df	CFI	IFI	AGFI	RMSEA
Recommended Value	<3.0	>0.90	>0.90	>0.80	<0.08
Measurement Value Indices	1.584	0.926	0.928	0.892	0.049
Structural Value Indices	1.838	0.900	0.902	0.870	0.059

**Table 3 ijerph-19-14173-t003:** Cronbach’s alpha, composite reliability, AVE, and correlations.

Constructs	Mean	SD	Cronbach’s Alpha	CR	AVE	SB	SP	PP	AD	HP	PC	PS
SB	2.74	1.11	0.823	0.833	0.625	**0.791**						
SP	3.34	1.04	0.708	0.807	0.583	0.354 **	**0.764**					
PP	2.55	1.03	0.700	0.824	0.610	0.125	0.126	**0.781**				
AD	2.81	1.00	0.753	0.826	0.543	0.292 **	0.340 **	0.107	**0.737**			
HP	3.65	0.96	0.847	0.844	0.644	−0.282 **	−0.136 *	−0.105	−0.251 **	**0.803**		
PC	3.24	1.00	0.828	0.838	0.565	−0.348 **	−0.316 **	−0.238 **	−0.295 **	0.489 **	**0.752**	
PS	3.45	0.89	0.719	0.803	0.577	−0.370 **	−0.078	−0.041	−0.096	0.296 **	0.191 **	**0.759**

Note: The square root of AVE is presented in bold. SB = school burnout; SP = social phobia; PP = parental phubbing; AD = SFV addiction; HP = happiness; PC = parent–child relationship quality; PS = perseverance. * *p* < 0.05; ** *p* < 0.01.

**Table 4 ijerph-19-14173-t004:** Loadings and cross-loadings.

Constructs	Items	PC	HP	AD	SB	PS	SP	PP
Parent–Child Relationship Quality	PC1	**0.778**	0.362	−0.052	−0.158	−0.003	−0.114	−0.047
	PC2	**0.790**	−0.017	−0.071	−0.046	0.160	−0.092	−0.054
	PC3	**0.717**	0.323	−0.077	−0.010	0.157	−0.230	−0.124
	PC4	**0.719**	0.178	−0.143	−0.218	−0.139	−0.049	−0.094
Happiness	HP1	0.351	**0.778**	−0.155	−0.041	0.122	0.032	−0.040
	HP2	0.225	**0.760**	−0.176	−0.167	0.100	−0.064	0.021
	HP3	0.096	**0.865**	−0.023	−0.117	0.179	−0.058	−0.052
SFV Addiction	AD1	−0.124	−0.041	**0.715**	0.028	−0.161	0.186	0.046
	AD2	0.016	−0.050	**0.775**	0.095	−0.157	0.062	−0.006
	AD3	−0.093	−0.114	**0.750**	−0.010	0.127	0.067	0.063
	AD4	−0.101	−0.099	**0.706**	0.247	0.082	0.078	−0.003
School Burnout	SB2	−0.145	−0.133	0.158	**0.733**	−0.272	0.214	0.043
	SB3	−0.223	−0.138	0.054	**0.810**	−0.136	0.119	0.079
	SB4	−0.020	−0.069	0.139	**0.826**	−0.121	0.144	0.012
Perseverance	PS1	0.011	−0.016	−0.031	−0.072	**0.779**	−0.082	−0.105
	PS2	0.111	0.137	−0.058	−0.197	**0.753**	−0.012	0.046
	PS3	0.023	0.314	0.004	−0.169	**0.746**	0.021	−0.015
Social Phobia	SP2	−0.181	−0.048	0.117	0.064	−0.018	**0.726**	−0.012
	SP3	−0.104	−0.024	0.127	0.128	−0.010	**0.817**	0.080
	SP4	−0.038	−0.019	0.096	0.188	−0.053	**0.744**	0.037
Parental Phubbing	PP1	−0.266	0.151	0.053	−0.032	−0.007	0.049	**0.731**
	PP2	−0.049	0.019	−0.026	0.081	−0.019	−0.024	**0.857**
	PP3	0.049	−0.298	0.081	0.060	−0.064	0.099	**0.749**

Note: The bold values are the indicator loadings of the main constructs, and non-bold values are the cross-loadings. All indicator loading values are significant (*p* < 0.001).

**Table 5 ijerph-19-14173-t005:** Common method bias test.

Constructs	Items	Substantive Factor Loading (R1)	R1^2^	Method Factor Loading (R2)	R2^2^
School Burnout	SB2	0.792	0.628	−0.025	0.001
	SB3	0.852	0.726	0.009	0.000
	SB4	0.938	0.880	0.014	0.000
Social Phobia	SB2	0.745	0.554	0.027	0.001
	SB3	0.853	0.727	0.087	0.008
	SB4	0.786	0.618	−0.124	0.015
Parent Phubbing	PP1	0.770	0.593	−0.064	0.004
	PP2	0.875	0.766	0.043	0.002
	PP3	0.724	0.524	0.077	0.006
SFV Addiction	AD1	0.723	0.523	−0.053	0.003
	AD2	0.798	0.636	0.018	0.000
	AD3	0.789	0.623	0.059	0.004
	AD4	0.724	0.524	−0.078	0.006
Happiness	HP1	0.875	0.766	−0.081	0.007
	HP2	0.820	0.673	0.040	0.002
	HP3	0.931	0.867	0.034	0.001
Parent–Child Relationship Quality	PC1	0.788	0.621	0.023	0.001
	PC2	0.801	0.642	−0.118	0.014
	PC3	0.818	0.670	0.128	0.016
	PC4	0.875	0.765	−0.049	0.002
Perseverance	PS1	0.837	0.701	−0.100	0.010
	PS2	0.733	0.537	−0.031	0.001
	PS3	0.823	0.677	0.135	0.018
Average		0.812	0.663	−0.001	0.005

**Table 6 ijerph-19-14173-t006:** Mediation test.

Consequences	Drivers	Bootstrapping	BC 95% CI		*p*-Value
		Indirect Effect	Lower Bound	Upper Bound	
Happiness	School Burnout	−0.078	−0.241	−0.001	0.047
	Social Phobia	−0.130	−0.295	−0.028	0.020
	Parental Phubbing	−0.016	−0.129	0.069	0.531
Parent–Child Relationship Quality	School Burnout	−0.099	−0.294	−0.002	0.041
	Social Phobia	−0.165	−0.378	−0.022	0.023
	Parental Phubbing	−0.020	−0.168	0.079	0.532
Perseverance	School Burnout	−0.036	−0.177	0.000	0.070
	Social Phobia	−0.060	−0.193	−0.003	0.032
	Parental Phubbing	−0.007	−0.074	0.027	0.354

Note: mediator: SFV addiction.

## Data Availability

The data used in this study are available on request from the corresponding author (chensijing@zust.edu.cn).

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
