# Peer review of "Drivers and Consequences of Short-Form Video (SFV) Addiction amongst Adolescents in China: Stress-Coping Theory Perspective"

_ijerph, 2022, doi:10.3390/ijerph192114173_

Round 1

Reviewer 1 Report

1.      The conceptual framework may be incomplete. In a typical SEM model examining a mediating relationship, an independent variable’s direct and indirect (via the mediator) effects on the outcome variable would be examined. The current study totally ignored the direct effects.

2.      If the conceptual framework is to be changed, the literature review section should reflect such a change.

3.      The current SEM did not control covariates. It is important to take into account demographics and factors that may have a confiding effect on the mediator and outcome variables.

4.      A little more specifics about data collection methods – for example, how were the participants recruited? 

Author Response

Reply to the first and second questions: Thank you for your valuable advice! It should examine a mediating relationship in a typical SEM model. Therefore, we measured the mediating relationships between the “drivers” of SFVs addiction and the “consequences” of SFVs addiction through SFVs addiction. The research model was modified accordingly, and the hypotheses about mediation relationships were also added in the Introduction part. For detail information, please refer to line 248 to line 287, line 297, line 427, and line 430.

The mediating analysis results showed that SFVs addiction appears to mediate the relation between school burnout and happiness, and parent-child relationship. Social phobia can negatively impact happiness, parent-child relationship, and perseverance through SFVs addiction. However, SFVs addiction failed to mediate the relationship between school burnout and perseverance. In addition, the mediation relationships between parental phubbing and happiness, parent-child relationship, and perseverance also failed. Discussion was also added in the Discussion part. For detailed information, please refer to line 495 to line 509.

Reply to the third question:  I agree with your advice. It is important to consider demographic factors that may have effects on the SFVs addiction and its consequences. Therefore, we included demographic factors of gender, grade, and watching time per day as control variables. The influence of control variables was discussed in the literature review section, from line 288 to line 295. The results showed that the control variables of gender (β = -296, p < 0.05) and watching time per day (β = 0.292, p < 0.001) influenced SFVs addiction significantly. Control variable of grade failed to impact SFVs addiction significantly. In addition, our results showed that all the three control variables had no significant impact on happiness, parent-child relationship, and perseverance. The results of the research model have also modified accordingly. For detailed information, please refer to line 297, line 428, line 400 to line 405.

Reply to the fourth question: This study adopted an online survey method to collect data. The questionnaire was made through Sojump (Wenjuanxing), a professional online questionnaire survey, examination, evaluation, and voting platform. Sojump has 2.6 million registered respondents with various demographic characteristics distributed across China. Sojump has been widely adopted to collect survey data under different research contexts, including e-commerce, air pollution, renewable energy, behavioral addiction. Sojump supports a variety of screening rules for invalid questionnaires. This study adopted Sojump to recruit participants. For detail description, please refer to line 301 to 308.

Reviewer 2 Report

The Article is generally well-written and interesting. However, it could be improved with the following changes.

- Some typos/words omitted make some of the sentences not completely grammatically correct. It is suggested for the manuscript to be checked.

TITLE

As the population sample is all located in China and is below 18, please add this in the title, for example:   Drivers and Consequences of Short-Form Videos (SFVs) Addiction AMONGST ADOLESCENT IN CHINA: A Stress-Coping Theory Perspective. 

ABSTRACT

Please add a sentence about the data analysis method in the abstract. Also, please add a sentence describing the population surveyed.

Instead of saying the theocratical implications are discussed, it is suggested to please put a sentence about the importance of the work and, who can benefit from it, what are the theoretical implications.

INTRODUCTION

When providing the definition of SFV, you should also make an example of the platforms on which such short videos are published (e.g. TikTok, Instagram., etc.).

The connection between SVF and hedonic technology usage should be better explained. Why do you think/have evidence that SVF are created for affective gratification? Have you investigated or reviewed the literature on motivation for creating SVF?

The statements below need a reference.

"SFVs are one type of such hedonic technology." - line 34

“Stress has been identified as a key predictive factor influencing behavioural addiction” - line 74

The following sentence is not totally clear; it is suggested to rewrite it:

“In addition, despite adolescents may encounter stress from various environments, a prior study has consistently treated stress as a general construct.” - line 80

PARTICIPANTS

The number at the beginning of a sentence should be in words; please replace the 22 with words in the following sentence:

“22 were removed due to (1) the adolescents having no 246 experience using SFVs apps;” - line 246

When reporting data about the participants, please use the number followed by the percentages throughout the whole section and the past tense as in the example below. Do this for the whole section.

"Among them, 83 (34%) WERE boys, and 159 (66%) WERE girls. Most adolescents WERE in high school (132 - 55%) and 62 (26%) in junior school."

MEASUREMENTS

Line 259 onwards - Please add a reference to the previous studies or explain better how the questionnaire was created/ extracted and the names of the questionnaires utilised to derive the current one. Please place the created questionnaire in an appendix to the paper or additional documents.

For the CFA please state which software / approach, was utilised and add a table with the values: χ2  , df,  χ2/df , χ2diff , GFI, RMSEA .

3.2 STRUCTURAL MODELLING

Please indicate which software was utilised for the structural equation modelling.

Author Response

Reply to Title: Thank you for your valuable advice. We have changed the title to “Drivers and Consequences of Short-form Videos (SFVs) Addiction Amongst Adolescent in China: A Stress-Coping Theory Perspective”.

Reply to Abstract:  We have added sentences describing the population surveyed and the data analysis: “The proposed model is tested based on the data collected from two hundred and forth two adolescents who are under the age of 18 and who have the experience watching SFVs from across China. Covariance-based structural equation modeling (CB-SEM) method is used for data analysis”. Please refer to line 18 to line 21.

We also described the importance of the work as well as the theoretical implications: “This study provides several theoretical implications. First, this study is one of the first to explain adolescents’ SFVs addiction from a stress-coping perspective, thereby enriching research in the field of SFVs addiction. Second, prior researches have rarely discussed the impact of stresses on addiction behavior from various environments in single research, therefore, contributes to the knowledge of stress-related research in SFVs addiction context. Finally, our study enhances our understanding of the impacts of SFVs addiction on its consequences, in both SFVs research context and social media research context”. Please refer to line 25 to line 31.

Reply to Introduction:  I agree with your opinion. We have given examples of the SFVs platforms when providing the definition of SFVs, such as TikTok, Instagram, Kuaishou, and Watermelon. Please refer to line 36.

In fact, we made a conclusion that SFVs are one of the subcategories of hedonic technology after reviewing the literature. For example, according to Zhang et al. (2019), SFVs provides personalized contents for users based on the analysis of their preferences. SFVs also offers fun stickers, video editing tools, and special effect filters to help users create funny videos. These customized contents and funny features have addictive hedonic value (Zhang et al., 2019). We have added the reason in the manuscript, please refer to line 44 to line 47.

Two references have been added for this statement. Please refer to line 89.

This sentence has been rewritten. Please refer to line 95.

We have replaced the number with words. Please refer to line 316 to line 319.

When conducting the descriptive analysis, we have used the number followed by the percentages throughout the whole section and used the past tense. For example, “among them, 83 (34%) were boys and 159 (66%) were girls. Most adolescents were in high school (132 - 55%) and almost 62 (26%) were in junior school. Furthermore, 103 (43%) watched SFVs for one to two hours, and 54 (22%) watched for two to four hours per day”. Please refer to line 319 to 328.

Reply to Measurements: All the measurements were adapted and modified from the prior study. For instance, four items were adopted from Salmela-Aro et al. (2009) to measure school burnout. Social phobia includes four items that were adapted from Peters et al. (2012). Parental phubbing was measured using three items adapted from Ding et al. (2018). SFVs addiction was measured using four items that were adapted from Chen (2019) and Choi and Lim (2016). Happiness involves three items that were adopted from Laffan et al. (2016). Four items were employed from Li and Zhou (2021) to measure parent-child relationship. Finally, perseverance was measured using three items employed from Santos et al. (2020). For detail information, please refer to line 333 to line 347. We have created questionnaire in the Appendix. Please refer to line 602.

A table of model fit indexes for both of the measurement model and structural model was added in the manuscript. Please refer to line 381.

The current study employed the covariance-based structural equation modeling (CB-SEM) using the AMOS (version 23.0) software to test our measurement and structural model. The advantages of AMOS were also described in the manuscript. Please refer to line 349 to line 355. 

Finally, we have read and modified the typos, and sentences that have grammar problems. For example, “which may decrease the capability to reduce excessive usage behavior automatically” was changed to “which may decrease the capability of reducing excessive usage behavior automatically”, please refer to line 129. “immersing themselves in SFVs enables them to escape from school stress as a coping strategy” was changed to “immersing themselves in SFVs is a coping strategy that can enable them to escape from school stress”, please refer to line 132.

Reviewer 3 Report

Dear Authors,

The topic of this study is of interest and potential value to the literature on addiction, social networks and adolescent well-being. However, I will only make one qualification.

1. Introduction

The introduction includes references that are not up to date. It would be interesting to establish the current state of the art. It is important to present studies with previous works in order to observe trends and possible changes in the last 5 to 7 years and not beyond, as there are a large number of references in the introduction below those years.

Bibliographic references starting from the introduction of the article are not in sequential order. For example, going from reference 3 (line 34) to references 5 and 6 (line 36), skipping reference 4 which appears later (line 40), and from references 13 and 14 (line 49) to 17 (line 54), skipping references 15 and 16 which appear later (line 70 and 154 respectively)..., the whole text should be reviewed according to the guide for authors in the references section:

  • References: References must be numbered in order of appearance in the text (including table captions and figure legends) and listed individually at the end of the manuscript. We recommend preparing the references with a bibliography software package, suchas EndNote, ReferenceManger or Zotero to avoid typing mistakes and suplicated references (…).

Similarly, throughout the introduction and some parts of the text authors are combined with the reference in square brackets, e.g. Cho et al. [30] (line 76), Huang et al. [4] (line 77), Huang et al. [4] (line 120), see lines 121, 166..., up to Sampasa-Kanyinga et al. [59] (line 399), the manuscript should be revised according to the authors' guidelines, and the phrase "various authors" may be recommended among other possible phrases to avoid referring to the author(s)....

In the text, reference numbers should be placed in square brackets [ ], and placed before the punctuation; for example [1], [1–3] or [1,3]. For embedded citations in the text with pagination, use both parentheses and brackets to indicate the reference number and page numbers; for example [5] (p. 10). or [6] (pp. 101–105).

2. Materials and Methods

It is important to distinguish between the method in terms of methodology with respect to the paradigm, the study design and the techniques and instruments to be used. It is not very correct to talk about the survey method, as it would be a technique used to collect information. This should be left out of section 2.1. Participants, as only the sample and how the sample was produced or chosen should be discussed in that section. 

2.1. Participants

The context in which the research takes place (which appears in the limitations) needs to be set out in more detail and the procedure for selecting the population until the final sample is obtained. There is also no comment on sample size and sampling error.

Finally, as mentioned in the first paragraph of this section, it is necessary to distinguish between the survey technique and the information collection instrument, the online questionnaire.

Let us recall the guide for authors:

They should be described with sufficient detail to allow others to replicate and build on published results. New methods and protocols should be described in detail while well-established methods can be briefly described and appropriately cited. Give the name and version of any software used and make clear whether computer code used is available. Include any pre-registration codes.

2.2. Measurement 

It would be useful to establish the criteria for reliability and validity of the questionnaire used, as well as a link to allow other researchers to access the questionnaire. A complementary file on the questionnaire used could also be included in this manuscript.

References

It is important to have adequate references, and these should be up to date and correspond to 50% of the last 5 years, the rest can be from no more than 5 years before, and by exception references from "classic" publications of more years are accepted (21 references in the last 5 years out of a total of 85 references, and by exception, from more than 10 years 29 references out of a total of 85 references).

On the other hand, reference 68 of Hofmann, W., Reinecke, L., Meier, A., Oliver, M. Of Sweet Temptations and Bitter Aftertaste: Self-Control as a Moderator of the 634 Effects of Media Use on Well-Being... (line 634) does not give the year of publication, the editors or coordinators of the book, the city and the publisher.

Author Response

Reply to Introduction: Thank you for your valuable advice! I agree with your opinion. Many references were out of date, thus, we have replaced the out of date references with references that were published in recent five to seven years. After modifying, there are 70 references in total, among them, 19 references were published before 2015, occupied 27% of the total references. It is worth mentioning that some of the references are the original ones that proposed theory. For example, the theory of distraction-conflict was presented by Baron in 1986. Stress-coping strategy was proposed by Lazarus in 1966. We should cite these original references in the manuscript. For detail information, please refer to the References part from line 606 to line 741.

For the reference of Hofmann et al., we added the year of publication which is 2017.

We have modified the bibliographic references in sequential order throughout the whole manuscript. Please refer to the manuscript.

We have avoided referring to the author(s) in the manuscript. For example, “Huang et al. [4] found that stress positively predicts SFVs addiction” was changed to “Stress was found to positively predict SFVs addiction [10]”. Please refer to line 134. “Geng et al. [46] and Niu et al. [47] have consistently discovered a significant relationship between parental phubbing and Internet addiction” was changed to “Research on Internet addiction [22,39] have consistently discovered that parental phubbing positively and significantly impacts Internet addiction”. Please refer to line 179. 

Reply to Materials and Methods: Thank you for your valuable advice! Survey technique and the information collection instrument, the online questionnaire is different. Therefore, we changed the sub-title of “2.1 Participant” to “2.1 Data Collection and Participant”. In the first paragraph of this section, we mainly described the survey technique, and in the second paragraph of this section, we mainly described the participants. Please refer to line 301 to line 328. In addition, determining the appropriate sample size is important for ensuring the quality of the study. Therefore, sample size and anticipated effect were articulated prior to the completion of the survey instrument. According to Westland (2010), the sample size for this study should reached to at least one hundred and seventy. There are 242 valid data in this study, thus, satisfying the sample size. Please refer to line 314.

We have created a questionnaire in the Appendix part. Responding references and specific measurements were shown in the Appendix. Please refer to line 602 to line 605.

Reviewer 4 Report

The article is clear, the methods and thesis properly statet. The conclusion are well thoughtful.

Author Response

Thank you for your comments!

Round 2

Reviewer 1 Report

The authors responded to the reviewers' comments and addressed the concerns. The paper is in good shape. 

Author Response

Thank you for your valuable comment.

Reviewer 3 Report

Dear authors,

The modifications made have improved the manuscript and it is ready for publication.

I congratulate you on your work.

Best regards

Author Response

Thank you for your valuable comment.